# Monolithic thin-film lithium niobate broadband spectrometer with one nanometre resolution

Giovanni Finco [1] ✉, Gaoyuan Li[1], David Pohl [1], Marc Reig Escalé [1], Andreas Maeder [1], Fabian Kaufmann [1] & Rachel Grange [1]

Miniaturised optical spectrometers are attractive due to their small footprint, low weight, robustness and stability even in harsh environments such as space or industrial facilities. We report on a stationary-wave integrated Fourier-transform spectrometer featuring a measured optical bandwidth of 325 nm and a theoretical spectral resolution of 1.2 nm. We fabricate and test on lithium niobate-on-insulator to take full advantage of the platform, namely electro-optic modulation, broad transparency range and the low optical loss achieved thanks to matured fabrication techniques. We use the electro-optic effect and develop innovative layouts to overcome the undersampling limitations and improve the spectral resolution, thus providing a framework to enhance the performance of all devices sharing the same working principle. With our work, we add another important element to the portfolio of integrated lithium-niobate optical devices as our spectrometer can be combined with multiple other building blocks to realise functional, monolithic and compact photonic integrated circuits.

Optical spectrometers are essential components in various fields of science and technology, from telecommunications to sensing and astronomy[1–4]. As for electronics, the integration of optical circuits is seeing rapid development, and miniaturised spectrometers are interesting for their compactness, robustness and stability. They represent a cost-effective solution to monitor relevant optical quantities even in environments with complicated access or where faults would make bulkier systems compromised and hard to replace[5]. Several implementations of such miniaturised systems have been proposed; they can be based on dispersive optics, tunable filtering, pattern recognition or Fourier-transform analysis[6]. The latter approach has its foundations in light interferometry, and it is accomplished either by micro-electro-mechanical systems[7–9], multiplexed or tunable Mach-Zehnder interferometers[10–14], or stationary-wave integrated Fourier-transform spectrometers (SWIFTSs)[15–21]. Integrated optical spectrometers are ideal candidates for replacing the bulkier counterparts and are especially useful for applications requiring long-time operations in harsh

environments, as faulty mechanical components would render the device unusable and irreparable.

SWIFTSs operate by locally sampling the intensity of a standing wave obtained by interference of counter-propagating optical modes in an integrated waveguide, either by reflection on a mirror (Lippman configuration) or by signal splitting and recombination. One powerful approach uses evanescent field scatterers (EFSs), which are patterned onto the waveguide in the form of nanowires[15,21], particles[18], or etched grooves[22]. They partially scatter out the wave to the near- and far-field; this produces an intensity pattern that is directly proportional to the optical energy, thus to the interferogram of recombined signals that can be visualised by means of an imaging system. The collected pattern can then be Fourier-transformed to the frequency domain in order to retrieve spectral information on the signal of interest.

One limiting factor to the available bandwidth is the requirement of single-mode waveguide operation, as a multi-mode behaviour would produce a difficult to interpret pattern. This would require

[1]ETH Zurich, Department of Physics, Institute for Quantum Electronics, Optical Nanomaterial Group, Auguste-Piccard-Hof, 1, 8093 Zurich, Switzerland.
✉e-mail: gfinco@phys.ethz.ch

sophisticated machine learning and pattern recognition algorithms to reliably retrieve the sought information. The arbitrariness of retrievable spectral shapes would, however, be compromised in this case. Single-mode operation is, thus, typically imposed, and on our platform, this can extend for hundreds of nanometres.

Moreover, SWIFTSs have the drawback of inherently under-sampling the waveform due to physical limitations on the dimension of the EFSs and their required separation in order to correctly sample the standing-wave at optical frequencies (i.e. standing-wave period and cross-talk between neighbouring samplers). The finite separation among the wires sets a lower bound to the minimum distance between samples of the interferogram, thus reducing the device bandwidth due to Nyquist criterion[15]. To overcome undersampling limitations, some approaches rely on thermo- or electro-optic modulation of the optical mode phase as it travels through the waveguide[14,21].

In this context, lithium niobate-on-insulator (LNOI) is an established electro-optic (EO) material platform enabled by the development of ion-slicing techniques[23]. Despite the challenges related to dry-etching of lithium niobate, great progress has been made over the years, and low-loss optical circuits of excellent quality can now be routinely manufactured[24–27]. LNOI is a progressively more used platform in the integrated optics community as it features all the advantages of bulk LN (second-order nonlinearity, high EO efficiency, broad transparency range and low optical loss) while also offering moderately high refractive index contrast necessary to waveguiding[28–31]. In particular, the strong EO material response enables efficient index tuning that can be exploited to actively change the phase of a travelling optical mode for applications in signal modulation[32–35], frequency-comb generation[36,37], radio-frequency to optical transduction[38,39], electric-field sensing[40,41] and also spectrometry[13,20–22].

Concerning SWIFTSs, electro-optic tuning of the interferometer arms allows for effective change in the relative path length between two counter-propagating signals, hence it conceptually corresponds to scanning the optical path difference (OPD) by moving one of the mirrors in traditional Fourier-transform infra-red spectrometry (FTIR) systems based on Michelson interferometers[20,21,42]. This approach allows to completely oversample the established standing-wave by carefully tuning the applied DC field to the modulation electrodes, such that the waveform is progressively shifted across the sampling region in order to overcome the physical limitations imposed by circuit and EFSs design. With this framework, at each voltage step, an image is taken corresponding to a snapshot of a different portion of the standing wave. Figure 1 illustrates a schematic of our devices, highlighting EFSs region and imaging system. Electrodes in push-pull configuration (to maximise the electro-optic efficiency) bias the interferometer via a DC voltage source to apply the phase delay. Scanning electron microscope (SEM) images show close-ups of EFSs, electrodes and waveguides cross-section, respectively.

At the current stage, detection relies on optical imaging with a table-top setup. A more effective solution would rely on micro-positioning of an array of InGaAs pixels on top of the scattering region for a fully integrated device. Alternatively, electrical readouts would be beneficial as they would not any more rely on optical imaging of the scattered pattern (which is inherently limited by diffraction). Recently, integrated photodetectors for SWIFTSs have been demonstrated[43], yet fabrication and contacting of hundreds of electro-optic nanosamplers is technologically very challenging.

When all data has been collected, images corresponding to different voltages are stitched together to reconstruct the complete, fully sampled waveform to be then Fourier-transformed. In 2020, our group demonstrated a high-bandwidth spectrometer based on a hybrid silicon nitride-LN platform (SiN-LN) by implementing a SWIFTS based on a closed-loop interferometer, focused ion beam (FIB)-deposited platinum nanosamplers, and EO effect on thin-film lithium niobate[21]. The proposed configuration made use of a loaded SiN ridge, deposited and

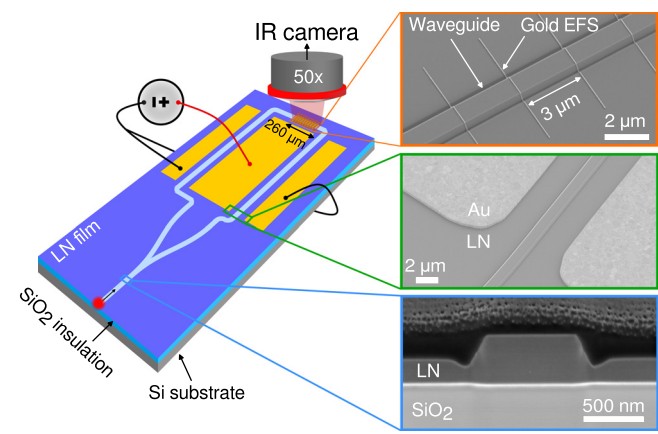

**Fig. 1 | Illustration of device and experimental setup.** SEM images of scattering gold nanowires patterned onto the waveguides, electrodes in the modulation region and FIB cross-section (covered in photoresist to protect against beam damage, stripped afterwards) of our waveguides are reported in the corresponding top, middle and bottom panels.

patterned for waveguiding on a LN film used for phase tuning. The reported spectral resolution of 5.5 nm at 1550 nm was limited by the sampled interferogram length due to the restricted field of view (FOV) of the imaging system. Here, we report on a fully monolithic configuration, lithographically fabricated, even for EFSs, to enable larger-scale production with high reproducibility and yield.

Monolithic implementation is, in general, preferred as it greatly simplifies the production of functional devices combining multiple building blocks on the same platform by minimising the fabrication steps. Furthermore, it allows for better exploitation of the material properties, such as enabling greater EO efficiency (due to higher optical-to-electrical mode overlap) and higher refractive index, which directly translates into an increased resolution due to better mode confinement and, thus larger effective index. Monolithic optical circuits on LNOI are becoming more compact, achieving bending radii as small as 40 μm when the LN film is fully etched[44], which is comparable to ultra-low loss Si-photonics[45]. Incidentally, optical losses are reduced by a factor of seven compared to hybrid SiN-LN configurations[21], and no optical mode transition is required when coupling the output of, e.g. an integrated source, to the device, thus maximising the sampled light intensity.

The spectral resolution of SWIFTSs is calculated similarly to that of FTIR systems, namely

$$\Delta\lambda = \frac{\lambda^2}{n_{eff}L},\qquad(1)$$

with $\lambda$ being the central wavelength of the measured signal, $n_{eff}$ its effective mode index and $L$ the sampled interferogram length. Ways of improving the device performance are based on increasing the effective mode index and extending the sampling region length.

Here, we propose two configurations to increase the spectral resolution of our SWIFTS relying on the monolithic nature, together with geometrical and symmetry considerations. We achieve a resolution improvement factor up to (nearly) five and demonstrate a framework to improve the resolution of SWIFTSs system down to the picometre range while maintaining the same experimental framework. We resolve continuous-wave (CW) laser spectra separated by 1.9 nm, and retrieve a full-width at half-maximum for single laser lines of 390 pm via fitting while retaining a broad bandwidth of operation. This breaks the trade-off barrier between the resolution and bandwidth of integrated spectrometers and can be applied to any SWIFTS device while maintaining a simple data acquisition procedure.

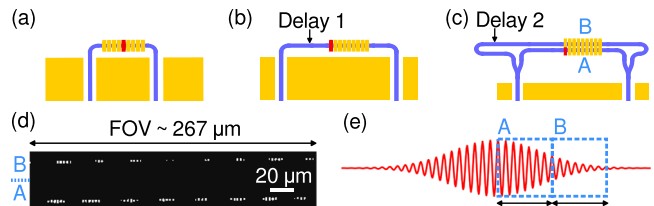

**Fig. 2 | Schematic drawing of the recombination and sampling region of our spectrometer devices. a** Symmetric, **b** asymmetric and **c** asymmetric with dual-scattering region configuration. Red lines in the scattering region mark the zOPD point. **d** Sample IR picture of a running experiment for design (**c**), where labels A and B indicate the corresponding interferometer branch where sampling of the standing wave occurs. **e** Qualitative representation of a standing wave where the sampled portions are highlighted by light-blue frames with the corresponding labels as in (**c**, **d**).

## Results

The standing wave in the recombination region of our circuits is symmetric about the zero optical path difference (zOPD) point. As such, the contained information is redundant and sampling the interferogram at both positive and negative optical path differences does not provide further insight into the spectral content of the signal. To gain a factor of two in spectral resolution, it is sufficient to break the symmetry of the circuit in order to shift the zOPD point to one end of the sampling region. This allows us to exploit the full FOV of our imaging system to retrieve useful information about the signal. Upon magnification, the FOV of the system is $267 \times 214\,\mu m^2$, and the diffraction-limited pixel size is approximately 420 nm.

Figure 2a, b shows the recombination region of a symmetric (a) and asymmetric (b) interferometer, where the red-marked scatterer indicates the zOPD position. The length mismatch between the two arms of the circuit is one scattering region long, thereby imparting a delay between the two signals that causes the zOPD to shift to one edge of the EFSs region. When the one-sided interferogram is reconstructed, it is then mirrored about the zOPD line in order to reconstruct the complete waveform from which the spectral content is retrieved by means of a fast Fourier transform. A second device configuration that further increases the spectral resolution is depicted in Fig. 2c: the input signals are split at the end of the modulation region by means of 50/50 Y-splitters; the two branches recombine after having travelled through arms of different lengths. The first branch travels along the same path as in the asymmetric design, while the second branch is designed with an additional, twice-as-long delay that allows for sampling of a different portion of the interferogram, thereby effectively extending the length of the sampled waveform. This overcomes the limitations given by the finite FOV. The two interferogram portions are separately reconstructed and then stitched together to build the extended waveform. Figure 2d, e shows a sample IR image as obtained from one of our devices, where the two parallel scattering regions are clearly visible, and a schematic of the working principle where the two sampled interferogram portions are highlighted by dashed frames.

Importantly, this framework can be expanded to four or even six parallel scattering regions, with the only limitation being the vertical separation between waveguides (that could introduce cross-talk among EFSs if too small) and the vertical size of the FOV. Every time the number of scattering regions is doubled with the corresponding signal delay, a factor of two in spectral resolution is gained.

### Design and fabrication
Waveguides are designed with a top width of 900 nm to extend the transverse-electric (TE) single-mode window of our circuits. They are patterned on a 600 nm-thick x-cut lithium niobate thin film, ion-sliced and wafer bonded on a silicon handle wafer with thermal oxide

isolation on top to provide the index contrast necessary for wave-guiding. The etching recipe produces trenches as deep as 20% of the target depth, which is 300 nm, hence an effective etch depth of 360 nm confines light into the waveguides[27]. With these design parameters, we calculate an effective mode index for the fundamental mode ($TE_0$) of $n_{eff} = 1.931$ at 1550 nm by using commercial software (Finite-difference eigenmode, Ansys Lumerical). Euler bends are used when waveguides need to turn in order to minimise bending loss and dispersion. Straight circuit sections-induced chromatic dispersion is compensated for in the spectral reconstruction step. Modulation electrodes and EFSs are patterned by electron-beam lithography and the lift-off process of gold. Scatterers are defined with a 3 μm spacing to avoid cross-talk, and a good image quality of 89 nanowires is obtained with the FOV on a single waveguide. Scatterers are patterned with a width $w_s = 50$ nm and thickness $t_s = 20$ nm in order to optimise the internal device efficiency. Coupling to the waveguide is done with the end-fire technique on FIB-polished input facets. SEM images of the relevant sample components are depicted in the right panels of Fig. 1. Importantly, the process is fully lithographic (except for facet preparation, which we do by FIB for convenience) in order to develop a fabrication flow that is suitable for mass-scale production with high yield.

The stitching voltage required to reconstruct the waveform (voltage needed to oversample the interferogram and completely avoid aliasing) depends on the modulation length of the interferometer, wavelength of the target signal and scatterers separation. Further EFSs engineering for increased scattering directionality, and reduced distance between samplers, together with an array of smaller pixels, would allow to minimise the power consumption of the device and bring the required modulation voltage to CMOS compatible ranges. These aspects will be the focus of further research.

### Optical characterisation
We test the resolution of our devices by coupling CW lasers around the telecommunication wavelength and reconstruct the spectrum using the three proposed designs. The theoretical resolutions of our three devices are $\Delta\lambda \sim 4.66$ nm (581 GHz, 19.40 cm⁻¹), 2.33 nm (290 GHz, 9.7 cm⁻¹) and 1.16 nm (145 GHz, 4.8 cm⁻¹) at 1550 nm, respectively for the symmetric, asymmetric and dual-scattering region configurations, as calculated using Eq. (1). The theoretical resolution of the dual-scattering region design is improved by a factor of nearly five compared to the hybrid SiN-LN version[21], owing to the increased mode index in monolithic LN and extended effective sampling length of the interferogram.

Figure 3 shows the resolved laser spectra fitted using a Lorentzian model, with the black solid line showing our reference obtained with a commercial optical spectrum analyser (OSA). The full-width at half-maximum (FWHM, $\delta\lambda$) of the fitted curves is used as a figure of merit for the quality of our designs and as a further quantification of the spectral resolution improvement. The obtained values for $\delta\lambda$ are 3.14, 1.16 and 0.39 nm, respectively. Coefficients of determination ($R^2$) of the Lorentzian fitted curves to data points increase from 0.93 to 0.99 for the three cases, which further confirms that the reconstructed spectra are approaching the actual CW laser line-shapes. It is important to highlight that the calculated FWHM should not be interpreted as the resolution of our devices since it is a quantity obtained via fitting of the data due to an insufficient number of points. The peak width could not be reliably calculated with just experimental points. It can, however, be used as a statistical measure to further testify to the improvement trend set by our framework.

CW lasers are also used to calibrate the scattering efficiency of nanowires and of the entire device, which we measure to be $\eta = 52\%$ (see Supplementary Material Note 1), and the stitching voltage required to fully sample the waveform, which in our case is around 15 V for a modulation length of 9 mm. The required voltage can be lowered by extending the electrode length in a linear fashion, bringing them closer to the waveguide, or reducing EFSs separation.

Next, we experimentally test the resolution of our devices by coupling two CW lasers at the same time and analyse the interferogram originated by the beating between their two frequencies. We successfully resolve the laser spectra with a minimum separation of 1.9 nm. Figure 4 shows the obtained results, where the inset displays measured and analytical interferograms obtained by considering effective mode indices of 1.9315 and 1.9310 for the shorter (1548.2 nm) and longer (1550.1 nm) wavelength, respectively. Data points are once again fitted to a (double) Lorentzian line-shape model (solid line), and the two laser peaks are well resolved, even though some artificial sidebands are generated during spectral reconstruction. These artefacts originate from DC drift, a known phenomenon occurring in LN[46]. It consists of a build-up of spatial charges, mostly accumulating at the waveguide cross-section edges, which shield the applied static field and effectively lower the electro-optic efficiency at low modulation speed. This results in a chirped intensity pattern versus applied voltage for each EFSs, which needs to be compensated for during data analysis (see Supplementary Material Note 2). Various approaches have been proposed to reduce the DC drift impact in LNOI devices operating at low speed and are mostly based on annealing of the samples in different atmospheres[38], chemical treatments[47] or cryogenic operation[48]. However, a permanent solution has yet to be found, and it remains an open question for the community. One solution to overcome this issue is to exploit thermo- instead of electro-optic modulation[49] or to modulate the signal at a relatively high speed (hundreds of kHz/few

MHz). The former approach is of less interest as it would increase power consumption, while EO tuning is notably more efficient. Our devices can be potentially operated at CMOS-compatible voltages with negligible power consumption by further engineering modulation and scattering section. Continuous calibration of the device can also be an option, requiring, however to substantially modify circuitry and experimental framework; for this reason, it was not attempted yet as simple operation was the main target of this study.

As a last step, we characterise the bandwidth of our spectrometers, focusing on the dual-scattering region design (Fig. 2c). Electro-optic tuning of our devices allows us to overcome the inherent under-sampling limitations given by the finite separation between EFSs, as a fine enough voltage step enables oversampling of the standing-wave and consequently eliminates aliasing and narrowing of the achievable bandwidth due to Nyquist criterion. The theoretical bandwidth is thus solely given by the single-mode condition of the designed waveguides, which feature an operating range extending from around 1.2 μm to far beyond 2 μm according to numerical simulations. Overall, the single-mode condition of our waveguides sets the lower limit, while the imaging system sets the upper limit to the bandwidth of our experiments, resulting in an operating range spanning from 1.2 to 1.7 μm (InGaAs pixel responsivity). Further discussion about the multi-mode operation and bandwidth limitations of our devices and experiments is provided in Supplementary Material Note 4, and it includes the analysis of interferograms originated by mode-mixing at shorter wavelengths.

We experimentally test the operating range by coupling CW laser lines between 1310 and 1635 nm and apply the same framework as described above to resolve the spectra and estimate the bandwidth. We successfully reconstruct the spectral features and thus measure a device bandwidth of at least 325 nm, as displayed in Fig. 5, where two bottom panels show zoom-ins of the two extreme laser spectra fitted again to a Lorentzian line-shape, giving FWHM of $\delta\lambda = 0.39$ and 0.63 nm, respectively with $R^2$ values of 0.95 and 0.98. Furthermore, to prove that our devices perform consistently across the measurable bandwidth, we sweep a tunable telecom laser in steps of 25 nm between 1460 nm and 1560 nm and reconstruct the spectra. We could retrieve each spectrum and follow the same data analysis framework described so far to calculate the FWHM via fitting. The bottom central panel displays $\delta\lambda$ as a function of wavelength, together with a linear fit (solid line) to the data. Fluctuations in the experimental values are to be attributed again to the random nature of the DC drift effect, which affects the width and position of the spectrum if not perfectly compensated for. The linear fit is reported as inspection of the reference spectra revealed an increasing trend of the FWHM at the sources themselves, and a comparison between measured and reference values is reported in Supplementary Material Note 4. Despite our devices overestimating the line-shape width by around one order of magnitude, the performance is consistent

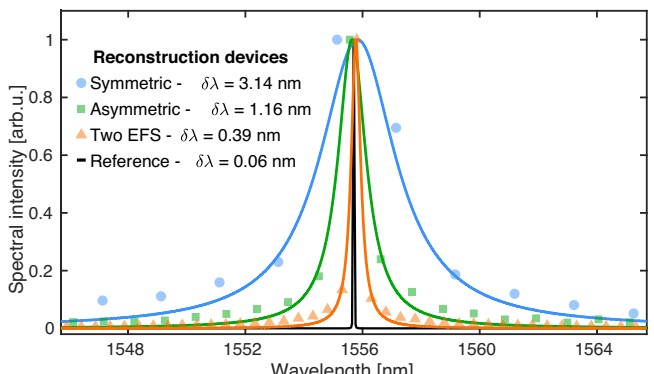

**Fig. 3 | Continuous wave laser spectra.** Reconstructed with our devices and experiments for a symmetric (light-blue, circular markers), asymmetric (green, square markers) and asymmetric with dual-scattering region (orange, triangular markers) configuration. Solid lines are Lorentzian fit to the respective data. The FWHM, as calculated from the Lorentzian fit of the data, are reported in the legend, while the black line shows a reference spectrum of the same light signal as obtained with a commercial OSA.

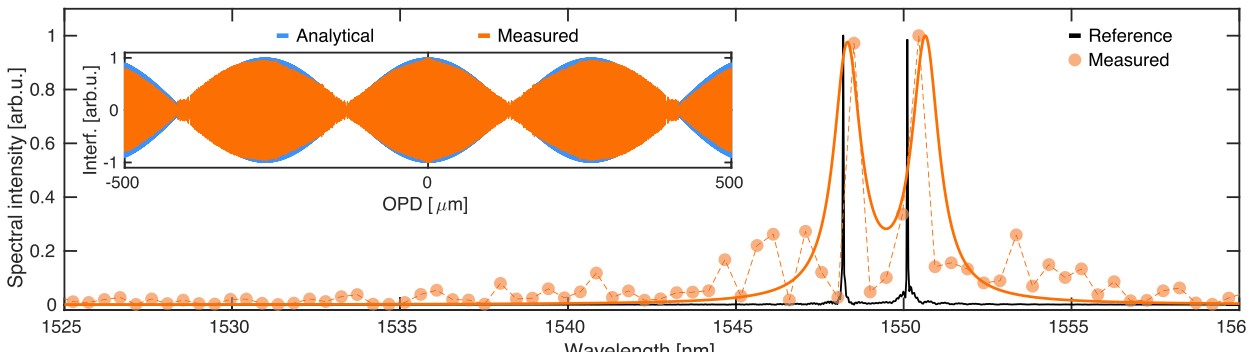

**Fig. 4 | Reconstructed spectrum for two neighbouring CW lasers.** Laser lines at 1548.2 and 1550.1 nm, respectively, are simultaneously coupled to our circuits. The solid line shows a fit of the datasets using a double Lorentzian line shape, while the inset displays the beating between the two frequencies as sampled by our device and reconstructed by assuming effective indices of the TE$_{01}$ mode of $n_{eff} = 1.9315$ and 1.9310 at a shorter and longer wavelength, respectively.

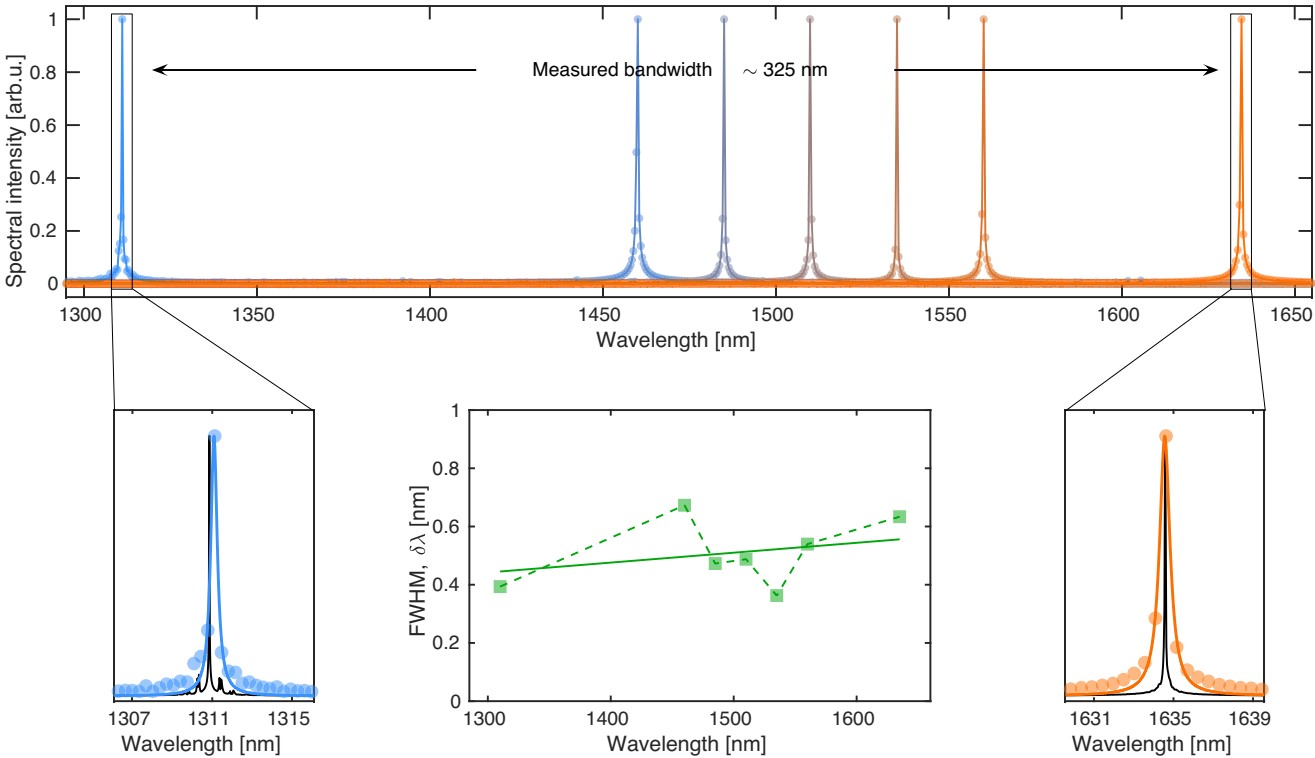

**Fig. 5 | Experimental measurement of our device bandwidth.** Obtained by coupling CW laser lines at 1310 and 1635 nm and applying the standard spectral reconstruction framework. We observe a minimum measured bandwidth of 325 nm, although the expected operating window is larger as measurements were limited by equipment availability. The central bottom panel illustrates the reconstructed FWHM, obtained by fitting the data as a function of wavelength together with a linear fit of the data points (solid line), to be compared to the reference FWHM measured with a commercial OSA (see Supplementary Material Note 4).

across the measurable bandwidth, and this further confirms the validity of our designs and framework.

## Discussion

We developed two innovative SWIFTSs designs, tested and fabricated on a monolithic LNOI platform, providing a framework to improve the spectral resolution of similar systems down to the sub-nm range. Our approach can indeed be scaled up, and further resolution improvement is possible with additional effort. Our best devices feature a theoretical spectral resolution of approximately 1.2 nm at 1550 nm and an optical bandwidth of at least 325 nm, mostly limited by the available equipment. By applying our framework and designing eight parallel recombination regions, a theoretical resolution of $\Delta\lambda = 290$ pm is achievable on our spectrometer while maintaining both available bandwidth and compact footprint. Concerning the latter aspect, at a fixed modulation region length (less than 1 cm in this case, that determines the required stitching voltage in a linear fashion), the footprint of our devices increases by roughly 30% for each resolution improvement factor of 2. Currently, a modulation length of 1 cm and a spectral resolution of $\Delta\lambda = 1.16$ nm correspond to a device extending over a 20 mm$^2$ area. The current design is, however, not optimised for compactness but rather to minimise potential issues due to bending loss and dispersion; further circuit design engineering would allow to reduce the device footprint further. The monolithic nature of the circuit allows for integration of the device with other components such as amplitude and phase modulators, and electro-optic frequency combs, already widely available on the LNOI platform.

From an experimental point of view, the resolution of our devices is currently limited by two factors: FOV of the experimental setup and DC drift effects in lithium niobate. The restricted FOV of the imaging system (number of pixels and need of a large objective magnification) limits the observable OPD and thus the achievable resolution, while drifting of the EO response causes neighbouring spectral features to merge when reconstructing the spectrum and thus prevents us from resolving some spectral shapes, especially when features are separated by a short spectral distance.

Relatively high-speed signal modulation and data acquisition would allow the elimination of the DC response drifting in lithium niobate; this will be a topic of further investigation. With this study, we nonetheless prove the suitability of the platform and demonstrate a framework that allows us to overcome the trade-off between bandwidth and resolution of SWIFTSs systems; importantly, this is done without the need to modify data acquisition or experimental techniques nor sacrificing device footprint or increasing circuit complexity.

On the other hand, complete miniaturisation of the device requires electrical readouts by means of heterogeneous integration or micro-positioning of InGaAs detectors array on top of the scattering region for fast data acquisition without the need for bulky experimental setups; this would also allow for extension of the observable OPD and thus improve the resolution further. Once on-chip detection is implemented, the device, combined with a broadband source, can be used to realise functional modules for real-world applications. For instance, spectrometric gas sensing in harsh environments like exhaust lines of vehicles or industrial facilities, where bulkier devices would not fit or survive, is one of our target applications. Similarly, astronomical spectroscopy would also benefit of the device thanks to the absence of moving parts, light weight, robustness and compact footprint.

## Methods

### Devices fabrication

Samples are fabricated on 20 × 20 mm$^2$ chips from an x-cut LNOI wafer with a 600 nm lithium niobate layer and 4.7 μm buried oxide layer thickness (NANOLN). Circuits are defined by means of electron beam

lithography (EBL - Raith Vistex EBPG 5200+) on a 650 nm-thick layer of hydrogen silsesquioxane-based flowable oxide (FOX16, from DuPont/Dow Corning) and then etched in an inductively-coupled plasma reactive-ion etcher (ICP-RIE - Oxford Instruments PlasmaPro Cobra 100) by argon ion milling with a process thoroughly described in[27]. Accurate etch-depth control is ensured by an interferometric end-pointing system, thus guaranteeing reproducibility between different samples. Redeposition and mask removal follow ICP etching; while annealing of the samples during 4 h at 500 °C allows to heal fabrication-related damages in order to reduce propagation losses. The next step is electrode deposition, patterned by EBL on a double-layer positive resist (polymethylmethacrylate, PMMA) and lift-off of 300 nm gold, similar to what is subsequently done for EFSs. A thin chromium layer (5 nm) is used to promote metal adhesion in both cases. Direct-laser writing and lift-off are then used to pattern metallic routes that connect the device electrodes to the edge of the chip, where they are then wire-bonded to a printed circuit board (PCB) for versatile control of the electro-optic tuning of each device. FIB (Thermofisher Helios 5 UX) milling with gallium ions is used to polish the input facets and prepare them for high-efficiency end-fire coupling.

## Measurements

We couple light to our circuits with the end-fire technique by means of lensed single-mode polarisation maintaining fibres (PMF, Oz Optics TPMJ-3A-1550-8). As light sources, we use tunable CW lasers (Keysight N7776C and Toptica CTL 1500/027048), two butterfly lasers (1064 and 1310 nm), and a super-luminescent diode (ThorLabs S5FC1005P). All sources are polarised and coupled to PMF; polarisation purity is ensured before starting the experiments. To image the scattered standing wave, we use a 50× infra-red objective (Mitutoyo 50× Plan Apo NIR, numerical aperture 0.65) with an InGaAs camera (Xenics Bobcat-640 GigE-1310) and a custom-built microscopy system. EO modulation of our circuits is done through a custom-designed PCB, that our samples are wire-bonded to. The signal is applied by means of a DC voltage source (Keysight E36106A) in steps of $\Delta V = 100$ mV. A custom software (based on C++ and PyMeasure) synchronises the IR camera and voltage source to take snapshots of the pattern at each step for a total of 10 frames per step in order to average out fluctuations and disturbances due to the environment. The camera background is subtracted from the averaged images before data processing. The interferogram retrieving and spectral reconstruction algorithms are written in MATLAB; they perform scattering efficiency and optical loss compensation, chirp correction and image stitching before Fourier-transforming the complete waveform. Reference spectra are obtained with a commercial OSA (ThorLabs OSA202C).

## Data availability

The data that support the findings of this study are available within the article and supplementary material. Raw datasets are available from the corresponding author upon request.

## Code availability

The codes to process the data subject of this study are available from the corresponding author upon request.

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

## Acknowledgements

We acknowledge support from the Scientific Centre of Optical and Electron Microscopy (ScopeM) and from the cleanroom facilities FIRST and BRNC of ETH Zurich and IBM Research Ruschlikon. Dr. Andrea Morandi and Dr. Alfonso Nardi from the Optical Nanomaterial Group at ETH Zurich for proofreading and validating the manuscript content. The project has been funded by the Swiss National Science Foundation (Project number 194693), the European Space Agency (Project number 4000137426) and the European Research Council (Project number 714837).

## Author contributions

G.F. designed and fabricated the devices; performed nanosamplers simulations, fabrication and test; performed data analysis and wrote the manuscript. G.L. optimised the experimental setup and data acquisition and conducted spectrometry experiments and data analysis. D.P. and M.R.E. developed the original device prototype, experimental and data analysis procedure. A.M. contributed to developing the designs, conducted simulations and designed PCB and electrical circuitry. F.K. assisted devices fabrication and developed etching and lithography processes. M.R.E. and R.G. conceived the idea of the project and acquired funding. R.G. supervised the project. All authors contributed to the discussion and paper revision.

## Competing interests

The authors declare no competing interests.
