## [Peer Review File · Nature Communications]

Monolithic thin-film lithium niobate broadband spectrometer
with one nanometre resolutionREVIEWER COMMENTS

Reviewer #1 (Remarks to the Author):

This work builds on work reported by the group in Nature Photonics in 2020. In that paper, they created a broadband hybrid thin-film lithium niobate–silicon nitride (LN-SiN) spectrometer with excellent spectral resolution. The advantages reported in the current paper include the following:

1. The nitride is eliminated from the structure and a full lithium niobate (LN) solution is presented.
2. The field of view is effectively improved by exploiting asymmetry in the interferometer.
3. Parallel scattering regions can be employed.

The key results are shown in the spectra, and I think the achieved spectral resolution is very nice. For this reason, I think the work is interesting and may attract many readers to the result.

My main concern is that the advantages of electro-optic (LN) compared to thermo-optic solutions for this application aren't so clear since high speed may not be very important here. The authors mention on page 4 (in the context of the bias drift issue) that thermo-optic solutions in LN itself may be considered but I suppose silicon could be used too. Are there any reasons other than reduced power consumption that motivate the use of LN then?

It's mentioned on page 2 that having a monolithic implementation is advantageous. I assume this is referring to having the LN etched waveguides rather than the LN-SiN structure instead, but it's a little unclear in the paper what advantages truly come from using LN etched waveguides. It somehow seems clear that the EO efficiency ought to be higher, but the other stated advantage of minimizing internal losses is unclear. Is the loss actually smaller here compared to the LN-SiN waveguides? Or are there also some advantages in terms of achievable miniaturization due to reduced bending radius in the all-LN structure?

There are a few typos in the paper (for instance "This aspects" on the top of page 4, some spacing and subscript errors in the references like in [45]) but they are all minor.

When mentioning the 2020 result on page 2, it would be helpful to readers to refer to [34] right there.

On page 2 it is mentioned that “Our approach instead minimises the number of electrical contacts” but I couldn’t understand how the number of contacts was reduced? Or does this just mean having a lot of integrated photodetectors necessitates a lot of electrical contacts?

Reviewer #2 (Remarks to the Author):

This paper proposes and demonstrates a hybrid Fourier transform spectrometer (FTS) on LNOI platform. It is hybrid in the sense that it introduces the technique of continuous tuning FTS to stationary-wave integrated Fourier-transform spectrometers (SWIFTS). By leveraging the electro-optic effect of Lithium Niobate to produce continuous OPD, the authors claim to solve the under-sampling problem of interferogram of conventional SWIFTS, which is the main novelty of this work. Another novelty lies in the novel layout design of the structure to increase resolution. The other parts, including theory, materials, main structure, calibration procedure are quite similar with previous demonstrations from the same group[1].

My following concerns need to be addressed before evaluation of the manuscript for publication at Nature communications.

1. One major concern is the value of introducing continuous OPD to SWIFTS. The authors criticize the under-sampling problem of SWIFTS, which I understand. Using continuous OPD can result in over-sampling of the interferogram, which I also understand. However, converting from under- to over-sampling should directly increase the bandwidth (according to classic Fourier transform theory, higher sampling rate gives a broader bandwidth after FT). But the bandwidth presented in this work (325nm) is much smaller than that of previously demonstrated SWIFTS (500nm) [1], on the same material platform from the same group, which confuses me. The authors attribute the bandwidth limitations to the equipment, but the same group has already experimentally demonstrated a 500nm bandwidth spectrometer. Why didn't they use the same equipment?

2. The authors emphasize the importance of low-cost integrated spectrometers and claim that this work can accompany larger scale production with high reproducibility and yield. However, I slightly disagree. The LNOI structure is relatively large compared to Si or SiN, and it is difficult for LNOI to bend to reduce the footprint. Additionally, the processable wafer size is much smaller. Moreover, this work requires the deposition of multiple gold nanowires with feature sizes as small as 20nm, which raises concerns about the scalability

and fabrication cost. It would be beneficial to provide concrete numbers regarding yield and cost to avoid misleading the readers.

3. The authors re-design the structural layout from symmetric to asymmetric, in order to shift the zero-OPD point to increase the resolution. When the one-sided interferogram is reconstructed, it is then mirrored about the zOPD line. This can only be valid when there is no fabrication variation between two arms, and the driving electrodes for two arms are exactly identical [2]. Have the authors checked these issues?

4. The experimental characterization in the main text is very weak, only reconstructions of a couple of laser peaks are provided. The resolution is characterized by the FWHM of the resolved peak and the bandwidth is claimed to be 325nm, which is the maximum separation between two laser peaks that can be reconstructed. In my personal viewpoint, both are unacceptable.

a) The resolution should be considered as the ability of the spectrometer to clearly resolve small features in a spectrum, thus both the FWHM of a single peak and the minimum separation between two peaks matter. In this work, the resolution should be modified to 1.9nm. One more question about the resolution, where does the 1.2nm spectral resolution (in the abstract) come from? The separation between peaks is 1.9nm, the narrowest FWHM of a single peak is 0.39nm. I see no experimental results related with 1.2nm resolution.

b) For the bandwidth, only two data points within 325nm are provided, which is inadequate in my viewpoint. Please provide more experimental results within 325nm optical range. Also, please plot the FWHM of the resolved peaks at different wavelengths, to see whether the spectrometer performs constantly within its bandwidth.

References:

[1] Pohl, D., Reig Escal´e, M., Madi, M., Kaufmann, F., Brotzer, P., Sergeyeve, A., Guldemann, B., Giaccari, P., Alberti, E., Meier, U., Grange, R.: An integrated broadband spectrometer on thin-film lithium niobate. *Nature Photonics* 14(1), 24–29 (2020)

[2] Souza, M.C.M.M., Grieco, A., Frateschi, N.C., Fainman, Y.: Fourier transform spectrometer on silicon with thermo-optic non-linearity and dispersion correction. *Nature Communications* 9(1), 665 (2018)

Reviewer #3 (Remarks to the Author):

The manuscript presents an important optimization development of miniaturized spectrometers based on the SWIFTS concept using the EO effect in LiNbO₃. A few years ago, the group has reported the main concept Ref. 21, where the optical path difference (OPD) can be adjusted by applying a voltage to the LiNbO₃ layer. A larger OPD is desired for higher spectral resolution, and continuous interferogram scanning is required to increase the bandwidth of the spectrometer. A commercially available InGaAs camera (640 x 512 pixels) serves as the photodetection system in the experiments described. In Ref.21, the authors symmetrically recorded the emitted light from one line of the samplers. In this study, the authors design the optical waveguide in a meander fashion and recorded only the necessary half of the interferogram. In this way, a longer OPD can be recorded with the camera, resulting in a ~5-fold increase in spectral resolution reaching 1 nm reflected in the title of the manuscript. The results are presented very clearly and the data are of a very high quality.

This reviewer considers the manuscript important for publication and has the following questions:

- Could authors more explicitly state what is the limiting factor for the spectral resolution in the reported spectrometer? Is it the resolution of the camera 640 x 512 pixels capturing only the area of 267x214 micrometers? Or is it the DC-drift in the EO effect?
- Could authors comment on the difference between the theoretically expected spectral resolution values (4.8, 2.4, and 1.2 nm) and experimentally measured FWHM values (3.14, 1.16, and 0.39 nm).
- Perhaps, the authors could state a scaling coefficient that links spectral resolution with the size of the spectrometer and another coefficient that links the size of the EO device to the OPD tunability range.
- How realistic is solving the DC-drift problem by switching to the MHz operation regime? In this case the camera also need to capture images with MHz frame rate. This would require very fast cameras and large intensities of light.

Reviewer #4 (Remarks to the Author):

This work demonstrated stationary-wave integrated Fourier-transform spectrometers (SWIFTSs) using monolithic thin-film lithium niobate.

My major concern is the novelty of this work. The same group published a very similar work on Nature Photonics before in 2020. Similarly electro-optic effect in LN had been used to overcome the bandwidth limitations imposed by undersampling. The major difference between this work and the NP work is that this work switched from hybrid thin-film lithium niobate–silicon nitride (LN-SiN) platform to lithium niobate-on-insulator. Besides, minor structural changes had been made e.g. asymmetric region configuration and usage of Y-splitter.

This work is incremental and seems not suitable for NC.

REVIEWER 1

Comment: *This work builds on work reported by the group in Nature Photonics in 2020. In that paper, they created a broadband hybrid thin-film lithium niobate–silicon nitride (LN-SiN) spectrometer with excellent spectral resolution. The advantages reported in the current paper include the following:*

1. *The nitride is eliminated from the structure and a full lithium niobate (LN) solution is presented.*
2. *The field of view is effectively improved by exploiting asymmetry in the interferometer.*
3. *Parallel scattering regions can be employed.*

The key results are shown in the spectra, and I think the achieved spectral resolution is very nice. For this reason, I think the work is interesting and may attract many readers to the result.

Response: First of all, we would like to thank the reviewer for the positive evaluation of the manuscript, below is a point-by-point response to the comments.

Comment: *My main concern is that the advantages of electro-optic (LN) compared to thermo-optic solutions for this application aren't so clear since high speed may not be very important here.*

Response: Thank you for pointing this out and allowing us to better clarify the concept. We agree that high speed operation is not the most important aspect, indeed realizing a fast device is currently not our main target. The reasoning behind the statement is that acquiring images at, for instance, 100 kHz would not be very complicated once an InGaAs detector array is placed on top of the scattering region, as the collection efficiency would greatly increase. The required integration time would then be substantially lower, hence allowing faster data acquisition and modulation speed offered by the electro-optic effect in LN. This regime of operation would allow to circumvent the DC drift issue.

Comment: *The authors mention on page 4 (in the context of the bias drift issue) that thermo-optic solutions in LN itself may be considered but I suppose silicon could be used too. Are there any reasons other than reduced power consumption that motivate the use of LN then?*

Response: The motivation to use LN is twofold:

1. First and foremost, the drastic reduction in consumed power which is of great importance when considering functional circuits where different building blocks operate synchronously and driven by the same electronic circuit. We are not there yet, but the device could operate at CMOS compatible voltages with further engineering, while thermo-optic tuning suffers of a much lower modulation efficiency due to power dissipation and would require a larger footprint to achieve the same amount of modulation by using the same voltage levels. Please refer to [1], which is about a device using TO modulation on silicon, and the device consumes a total of 5 W of electrical power to achieve full modulation.
2. LNOI is becoming a progressively more mature platform with devices being sold commercially from different companies. As of now, mostly ultrafast electro-optic modulators are on the market, but frequency-combs and other devices have been reported and can be operated very efficiently in terms of consumed power (see for instance [2], a 25 GHz EO-comb can be driven with only 0.68 W of electrical power). Our goal is to eventually combine our spectrometer with other components in LNOI to realize a functional device (e.g. a gas sensor using an EO comb and an absorption section to then detect the gas) that may find applications in space or industrial environments. In this context, it is important to use the same platform in order to minimize loss of optical power (due to e.g. optical interconnects between different platforms), have a robust, simple and reliable fabrication, and operate the device efficiently in terms of electrical power. As of today, no monolithically integrated spectrometer on LNOI was demonstrated; we show it and provide a way of reaching, in our opinion, remarkable resolution.

Thank you for giving us the opportunity of better clarifying this aspect, we added the sentence:

The former approach is of less interest as it would increase power consumption, while EO tuning is notably more efficient. Our devices can be potentially operated at CMOS compatible voltages with negligible power consumption by further engineering modulation and scattering section.

to better clarify point 1., while point 2. is addressed on page 2 (see below).

Comment: *It's mentioned on page 2 that having a monolithic implementation is advantageous. I assume this is referring to having the LN etched waveguides rather than the LN-SiN structure instead, but it's a little unclear in the paper what advantages truly come from using LN etched waveguides. It somehow seems clear that the EO efficiency ought to be higher, but the other stated advantage of minimizing internal losses is unclear. Is the loss actually smaller here compared to the LN-SiN waveguides? Or are there also some advantages in terms of achievable miniaturization due to reduced bending radius in the all-LN structure?*

Response: This questions find partial answer in our previous response, as monolithic implementation allows for easier, scalable, robust and reliable fabrication of functional circuits within the same platform. EO efficiency is also higher indeed due to the optical mode being fully confined in the lithium niobate waveguide. One example where this is highly beneficial is our target application, that is a larger-scale device combining multiple building blocks, where the output of a monolithic LN circuit acting as source (e.g. EO comb) needs to be coupled to our spectrometer. In case the latter is realized with a SiN loaded ridge waveguide (as in our proof-of-concept in 2020), mode conversion would be required from the monolithic LN circuit. This process would be inevitably lossy, hence causing an already weak signal to degrade further. With a monolithic implementation this problem is avoided and coupling efficiency to the device can be maximised.

Regarding propagation losses, the monolithic LNOI waveguides have seven times lower loss than the SiN-LN counterpart. Referring to our previous work [3], in the supplementary material it is shown that propagation losses were as high as 1.4 dB/cm, while our current layout features 0.2 dB/cm loss, in line with the state-of-the-art for LNOI technology [4, 5]. One of the main sources of loss in SiN-loaded waveguides at the telecom C-band comes from Si-H bonds absorption, and eliminating them requires delicate fabrication steps that are not always successful (depending also on how SiN is deposited). These include high temperature annealing that may result on thin film cracking due to stress differences between materials and/or adhesion issues between layers.

The most advanced approach to implement hybrid SiN-LN circuits with comparable optical performance is the photonic Damascene process recently demonstrated for the LNOI platform [6]. This approach is very involved, fabrication is complicated and does not suit our purposes which aim at a simple and easily scalable manufacturing fully on LNOI.

Focusing instead on circuit miniaturisation, at the current stage of our technology the default bending radius featuring low optical loss is 80 μm ; however, it has been recently shown that full etch of the LN film allows to bend waveguides with a radius as small as 40 μm while retaining low propagation loss [8]. Hence, further miniaturisation of the circuit is possible and would go well beyond what can be accomplished in a SiN-LN hybrid platform (while retaining all the advantages of LN). Fully etching the thin-film can result in a lower EO efficiency, thus a trade-off is to be found in that regard.

We modified the relevant paragraph to:

Monolithic implementation is in general preferred as it greatly simplifies the fabrication of functional devices combining multiple building blocks on the same platform by minimising the fabrication steps. Optical circuits on LNOI can be more compact, achieving bending radii as small as 40 μm when the LN film is fully etched [43], which is comparable to ultra-low loss Si-photonics [44]. Furthermore, it allows to better exploit the material properties such as enabling greater EO efficiency (due to higher optical-to-electrical mode overlap) and higher refractive index which directly translates in an increased resolution due to better mode confinement and thus larger effective index. Incidentally, optical losses are reduced by a factor of seven compared to hybrid SiN-LN configurations [21] and no optical mode transition is required when coupling the output of, e.g. an integrated source, to the device, thus maximising the sampled light intensity.

and moved the sentences:

We achieve a resolution improvement factor up to (nearly) five and demonstrate a framework to improve the resolution of SWIFTSs system down to the picometre range while maintaining the same experimental framework. This breaks the trade-off barrier between resolution and bandwidth of integrated spectrometers, and can be applied to any SWIFTS device while maintaining a simple data acquisition procedure.

to the next paragraph, after introducing the resolution equation. We hope this reworking of the section makes the discussion clearer for a broader audience, thank you for pointing it out.

Comment: *There are a few typos in the paper (for instance “This aspects” on the top of page 4, some spacing and subscript errors in the references like in [45]) but they are all minor.*

Response: Thank you for highlighting this, we checked the document again several times and tried to remove all typos and mistakes, but please let us know should there be more.

Comment: *When mentioning the 2020 result on page 2, it would be helpful to readers to refer to [34] right there.*

Response: Thank you for the suggestion, we added reference [21] after the sentence ([34] is an electro-optic modulator, we believe it being not that relevant after such a sentence, or was the suggestion to add [34] intentional?).

Comment: *On page 2 it is mentioned that “Our approach instead minimises the number of electrical contacts” but I couldn’t understand how the number of contacts was reduced? Or does this just mean having a lot of integrated photodetectors necessitates a lot of electrical contacts?*

Response: The sentence meant indeed that having a lot of photodetectors requires a large amount of electrical contacts while taking images of the scattered pattern reduced the number drastically. Eventually we believe that the most suitable approach would be precise positioning of an array of commercially available and packaged InGaAs pixels on top of the scattering region; that would not require independent contacting of hundreds of nanosamplers. We removed the sentence from the text as we agree that it was misplaced, and re-wrote the paragraph to make it clearer.

REVIEWER 2

Comment: *This paper proposes and demonstrates a hybrid Fourier transform spectrometer (FTS) on LNOI platform. It is hybrid in the sense that it introduces the technique of continuous tuning FTS to stationary-wave integrated Fourier-transform spectrometers (SWIFTS). By leveraging the electro-optic effect of Lithium Niobate to produce continuous OPD, the authors claim to solve the under-sampling problem of interferogram of conventional SWIFTS, which is the main novelty of this work. Another novelty lies in the novel layout design of the structure to increase resolution. The other parts, including theory, materials, main structure, calibration procedure are quite similar with previous demonstrations from the same group [1]. My following concerns need to be addressed before evaluation of the manuscript for publication at Nature communications.*

Response: We thank the reviewer for their valuable comments and for raising concerns that helped us improving the quality of our work, below is a point-by-point response to the addressed aspects.

Comment: *One major concern is the value of introducing continuous OPD to SWIFTS. The authors criticize the under-sampling problem of SWIFTS, which I understand. Using continuous OPD can result in over-sampling of the interferogram, which I also understand. However, converting from under- to over-sampling should directly increase the bandwidth (according to classic Fourier transform theory, higher sampling rate gives a broader bandwidth after FT). But the bandwidth presented in this work (325nm) is much smaller than that of previously demonstrated SWIFTS (500nm) [1], on the same material platform from the same group, which confuses me. The authors attribute the bandwidth limitations to the equipment, but the same group has already experimentally demonstrated a 500nm bandwidth spectrometer. Why didn't they use the same equipment?*

Response: Indeed, we also expect a broader bandwidth for our devices, yet we would like to highlight some differences between this and the previous work that hopefully will help clarify our results. The goal of our current research is to develop a functional gas sensing module that uses an integrated spectrometer to detect molecular fingerprints. To do this, we decided to shift the operational bandwidth of our devices towards longer wavelengths (NIR is already a non-optimal spectral window for gas detection, but certainly the longer the wavelength, the better). The bandwidth of our devices is limited only by the single-mode condition of the designed waveguides, which as we stated in the manuscript starts around 1250 nm and extend to more than 2000 nm. It is important to mention that the waveguides could be re-designed to operate at shorter wavelengths, thanks to the broad transparency of LN, but this would conflict with the ultimate goal of our research. We do agree and realise that, to show the full potential of the device, a design operating entirely within the wavelength range we can currently measure would have been the best approach. However, we preferred to directly implement a design suitable for our long-term goal and rely on our previous results to comment on its full operating range.

We did make use of the same equipment, which indeed sets an upper limit to the detectable wavelength at around 1700 nm due to the responsivity of InGaAs camera cutting off in that range. Regarding the lower wavelength limit, even though our camera can detect shorter wavelengths, the waveguide is not single-mode any more below 1250 nm, which is one of the differences compared to the previous version on SiN-LN (that used a different waveguide design). Hence, it is not easy to prove a broader bandwidth with the equipment we have available as we already pushed the tunable telecom lasers to their limit and used one butterfly laser at 1310 nm. Specifically, at 1064 nm the waveguide supports up to the third order TE mode, which has strong coupling to the fundamental TE mode due to their same parity, and this limits the device bandwidth because of mode-mixing. We would also like to emphasize that, when talking about bandwidth (page 5), we specify that the equipment limits the bandwidth of the experiment, not of the device. Reliable cameras and sources operating above 1700 nm are very hard to find, expensive and we unfortunately do not have any available to prove the operation at the long wavelength edge of the single-mode window. We realised that the statement might not have been sufficiently clear, so we rephrased it (see below) to highlight that the single-mode condition sets the lower limit, while the experimental setup and available equipment sets the upper limit to the measurable bandwidth.

To support the above statements, we report in Fig. 1 the intensity trace obtained by monitoring a single scattering nanowire versus applied modulation voltage for two cases: multi-mode ($\lambda = 1064$ nm, blue

line) and single-mode ($\lambda = 1635$ nm, orange line) operation. We strived to couple and reconstruct a CW laser spectrum at $\lambda = 1064$ nm with the same procedure, equipment and algorithm, however this was not feasible without substantially increasing the data analysis routine complexity. By inspecting the measured patterns it is clear that for a waveguide supporting multiple optical modes our procedure suffers of mode-mixing. Indeed, higher order modes having a lower effective index interfere by producing a standing wave of different periodicity with respect to the fundamental one. Overlaps between the interferograms results in a hard to interpret pattern that cannot be directly related to a CW laser interferogram. As anticipated in the main text, multi-mode operation would require complicated pattern recognition and deconvolution algorithms in order to separate the contributions of different optical modes, which is beyond the scope of this work.

Figure 1: Comparison of the interferogram trace versus applied modulation voltage for multi-mode ($\lambda = 1064$ nm, blue line) and single-mode ($\lambda = 1635$ nm, orange line) operation. The trace is obtained by monitoring the intensity profile from a single scattering nanowire and highlights the difficulty of reconstructing the CW interferogram in case of mode mixing.

To further support the multi-mode operation statement, Fig. 2 shows the simulated transverse-electric optical modes in our circuits at $\lambda = 1064$ nm, 1310 nm, 1550 nm and 1635 nm, from which it is clear how the waveguide supports up to the third order mode at $\lambda = 1064$ nm. Even though it is weakly guided, the TE_{03} mode has a large overlap with the fundamental one, which leads us to believe that mode-mixing is occurring in our waveguides and results in the interference pattern reported in Fig. 1. To conclude, the bottom panel of Fig. 2 reports the dispersion curve of the fundamental TE mode as obtained with our simulations, with the vertical dashed lines indicating the wavelengths corresponding to the reported mode-profiles.

We thank the reviewer to give us the opportunity of better clarifying this aspect and confirm our statements through more thorough experimental characterisation, we have included a discussion on multi-mode operation and bandwidth limitations of our experiments in a new Supplementary Material section (SM4). We hope that this solves the reviewer's doubts. We slightly re-phrased the paragraph and a sentence has been added on page 5 of the main text to point the readers to the relevant discussion in the Supplementary Material:

Overall, the single-mode condition of our waveguides sets the lower limit, while the imaging system sets the upper limit to the bandwidth of our experiments, resulting in an operating range spanning from 1.2 to 1.7 μm (InGaAs pixel responsivity). Further discussion about multi-mode operation and bandwidth limitations of our devices and experiments is provided in the Supplementary Material SM4, and it includes the analysis of interferograms originated by mode-mixing at shorter wavelengths.

Comment: *The authors emphasize the importance of low-cost integrated spectrometers and claim that this work can accompany larger scale production with high reproducibility and yield. However, I slightly disagree. The LNOI structure is relatively large compared to Si or SiN, and it is difficult for LNOI to bend to reduce the footprint.*

Response: We agree and there is no doubt that Si photonics can achieve more compact circuits and higher densities. Nevertheless, we would like to highlight that LNOI is getting more and more

Figure 2: Simulated transverse-electric optical modes at the wavelengths of $\lambda = 1064$ nm, 1310 nm, 1550 nm and 1635 nm in our circuits. At 1064 nm the waveguide support up to the third order mode (although weakly guided), which has a large overlap with the target fundamental TE mode. The single-mode window is calculated to start around 1200 nm. Effective indices and mode areas are reported in the respective panels, while the bottom plot shows the calculated dispersion curve of the fundamental TE mode versus wavelength.

mature and is approaching comparable dimensions. We would like to refer to the following recent paper [7] where it is shown that low-loss silicon photonics requires bending radii in the range of 40-60 μm . Similar arguments apply to SiN. Indeed, in our opinion footprints can be compared when optical losses are comparable too, especially for fairly large devices. A recent publication [8] showed that full-etch of LN films allows to reach bending radii in the range of 40 μm as well while retaining low propagation loss. We agree that Si-based photonic circuits have a smaller footprint, yet the gap between platforms is being reduced as the technology advances. Moreover, tuning the OPD requires electrodes that increase in length as the desired voltage to be applied (or power consumption in general) gets smaller. Electrodes indeed constitute the majority of the device footprint, and this applies to Si or SiN as well, with LNOI having the advantage of an inherently reduced power consumption thanks to the electro-optic effect being more energy-efficient than thermo-optic tuning. Furthermore, insertion loss for phase shifters in LNOI is much lower than in Si where carrier injection worsen the optical performance. Considering [1], the modulation region covers an area that is approximately 20 times smaller than our device, yet running the device requires 5 W of electrical power due to thermal dissipation. It is hard to compare electrical power consumptions between the two cases as our device works with a capacitive modulation, but not requiring constant supply of electrical power, electro-optic modulation is many orders of magnitude more efficient.

Comment: *Additionally, the processable wafer size is much smaller. Moreover, this work requires the deposition of multiple gold nanowires with feature sizes as small as 20 nm, which raises concerns about the scalability and fabrication cost. It would be beneficial to provide concrete numbers regarding yield and cost to avoid misleading the readers.*

Response: We understand the concern and agree that the LNOI platform is not yet ready for production of samples on the same scale as Si/SOI/SiN where 8" wafers and even larger are routinely processed. There is no doubt about the comparison between maturity of the two platforms. However, we would like to point out that 6" LNOI wafers are available and used to produce devices of different kinds with

multi-project runs (see for instance Centre Suisse d'Electronique et de Microtechnique, CSEM). Even though at the current stage we employ electron-beam lithography to pattern our circuits, and such a tool is not suitable for mass-scale production, all the features we consider can be patterned by means of deep-UV lithography (DUV). The concern about the possibility of patterning metallic nanowires with features as small as 50 nm (in-plane size, not 20 nm which is the out-of-plane thickness) is understandable, yet DUV systems with ArF excimer lasers (193 nm, relatively common DUV tools) have been shown to reach sub-100 nm features with moderately low effort, with or without water immersion (see for instance the TWINSCAN line from ASML, models XT:1460K/NXT:870/XT:860N/XT860:M <https://www.asml.com/en/products/duv-lithography-systems>). We would also like to stress that we did not conduct thorough optimization on the largest nanowire size that still allows to conduct reliable experiments, and it is possible that larger features would still suit the purpose. Moreover, at a foundry level, nanowires would not be manufactured by means of lift-off process but rather with lithography and etch of a metallic film, which simplifies the process on a large-scale. Incidentally, according to our simulations such scattering elements can also be made out of amorphous silicon.

In our case, we can easily realise 8 devices on a $2 \times 2 \text{ cm}^2$ chip, and out of a 6" wafer one can get 32 chips of such size, totalling up to around 260 devices on a single wafer. This without considering that our chip templates include buffer regions for handling and aligning during lithography steps. For a full-wafer process, the number of manufacturable devices increases by about 30%. Our fabrication process (EBL and lift-off) shows a yield very close to 100%, with high reproducibility in device performance. We would like to emphasize that we are working at a pure research level and optimising costs for large-scale production is beyond the scope of our research. Nevertheless, wafer-scale production with minimised processing steps, which is what we strived to achieve in our work, goes hand-in-hand with manufacturing cost reduction.

Comment: *The authors re-design the structural layout from symmetric to asymmetric, in order to shift the zero-OPD point to increase the resolution. When the one-sided interferogram is reconstructed, it is then mirrored about the zOPD line. This can only be valid when there is no fabrication variation between two arms, and the driving electrodes for two arms are exactly identical [2]. Have the authors checked these issues?*

Response: We also had this concern for very long time indeed and put a lot of effort to understand to what extent fabrication imperfections were affecting our devices performance. To support the reproducibility of our fabrication workflow, we would like to highlight our recent publication [5] where, in section 3.5, we discuss reproducibility of our optical measurements by analysing transmission spectra and quality factors of racetrack resonators of varying length fabricated on many different chips during different days. We could prove close to perfect reproducibility as confirmed by figure 8 in the manuscript, and we believe this should confirm that our fabrication process is highly reliable. Regarding variations within a single device, we always include labels and reference points in proximity of relevant spots in the scattering region of our spectrometer as shown in Fig. 3. This is done to verify that features such the zOPD line falls at the designed location. Furthermore, we always include buffer regions with additional scattering nanowires to further check that asymmetric designs perform as they should. We always find perfect match between designed circuit and expected location of zOPD by coupling a broadband source (narrow interferogram, where the point of maximum brightness is clearly visible) to our circuits. Such a point always falls in close proximity (hundreds on nanometres) to the designed scatterers, hence within experimental setup limitations in terms of camera resolution.

Regarding the driving electrodes, the deposition is done by lithography and lift-off process, which implies the same metal quality and feature size. The push-pull configuration enables more efficient tuning, but does not affect the shape of the interferogram as it only acts by shifting it back and forth across the circuit. As for the geometry of the circuit, please refer to the following atomic force microscope (AFM) scan of the modulation region profile in Fig. 4 for an indicative measurement, keeping in mind that an AFM is meant to measure heights and roughnesses rather than widths and distances. Features height is indeed exactly identical as expected. From such a scan we could measure a maximum electrode-to-waveguide offset difference of few tens of nm, which is probably due to measurement inaccuracy as patterning of the circuit is done fully with EBL. The only issue that may be introduced by non symmetric electrode geometry is an unbalanced metal-induced optical loss, which on such a length scale (waveguide-metal distance) is irrelevant. Please ignore the bumps on the right side of left electrode

Figure 3: Highlight of the circuit layout corresponding to the scattering regions for an asymmetric and dual-scattering region device as designed in GDS format. The scattering regions are labelled and marked with reference elements in order to mark the expected position of the zOPD and frame the field of view of the imaging system for aligning the experiment.

and waveguide, which are artefacts introduced by the AFM during the re-scan.

Figure 4: Atomic force microscope profile of electro-optic modulation region, showing the accuracy of electrodes patterning and deposition.

The major contribution in terms of incorrect standing-wave sampling is to be attributed to the scattering nanowires, which are more prone and susceptible to fabrication tolerances. This is compensated for during the calibration step by measuring CW laser interferograms (that have a known waveform), and correcting for potentially different scattering efficiencies of different nanowires. This map is then stored and applied to renormalise the data.

Comment: *The experimental characterization in the main text is very weak, only reconstructions of a couple of laser peaks are provided. The resolution is characterized by the FWHM of the resolved peak and the bandwidth is claimed to be 325nm, which is the maximum separation between two laser peaks that can be reconstructed. In my personal viewpoint, both are unacceptable.*

Response: We have added several measurements of CW lasers within the measurable bandwidth and discuss them below. Please note that we also show a reconstructed broadband source spectrum in the Supplementary Material section SM3. This data was deliberately left out of the main text as we wanted to focus on resolution improvement, which is not clearly illustrated by such an experiment.

Comment: *a) The resolution should be considered as the ability of the spectrometer to clearly resolve small features in a spectrum, thus both the FWHM of a single peak and the minimum separation between two peaks matter. In this work, the resolution should be modified to 1.9nm. One more question about the resolution, where does the 1.2nm spectral resolution (in the abstract) come from? The separation between peaks is 1.9nm, the narrowest FWHM of a single peak is 0.39nm. I see no experimental results related with 1.2nm resolution.*

Response: We agree that, from an experimental point of view, the smallest detectable separation between narrow features is one way of measuring the devices resolution, similar to what is done with

imaging systems limited by diffraction. Nevertheless, we would like to point out that neither of the cited papers used this approach for their reported resolution (Souza et al. seem to report on the distance between data-points in the reconstructed spectrum - there is no clear statement -, Pohl et al. use the width of the reconstructed CW laser peak to quantify the resolution). The 1.2 nm resolution we report is the theoretical value (please note that the fact that this is the theoretical, not experimental, resolution was already specified in the manuscript) that would be achieved in absence of drifting effects, and that can be calculated using eq. (1) in the manuscript, as it is the only quantity that depends on design parameters and can be calculated a priori. We tried also to retrieve laser peaks with such a spectral separation; the spectral features were not as clear as for the 1.9 nm even though two features could be distinguished anyway. We decided to exclude those measurements from the manuscript and first find a way to reduce or eliminate DC drift effects.

Please note that we did not use the FWHM of the reconstructed spectra as a measure of device resolution. The FWHM is calculated from a fit to the experimental data. Datasets do not include points at half the peak height (nor are they available at the same height), therefore it is not possible to use them to calculate the FWHM consistently and fitting is required. We reported the theoretical resolutions as calculated via eq. (1) in the manuscript as it is the only quantity that can be directly related to design parameters. Using the distance between points in the reconstructed spectrum is not a reliable approach as, in principle, one could increase their density by applying a spline interpolation to the data. To further confirm that our designs introduce a clear performance improvement for our devices, we used a fitting approach to perform a statistical analysis on the reconstructed datasets (indeed the coefficient of determination for fitting to a Lorentzian line-shape consistently increases as the effective sampling region gets longer). However, the FWHM obtained by the fit should not be interpreted as the devices resolution but just a figure of merit to confirm the validity of our approach.

We expanded the abstract to clarify the distinction between theoretical and measured quantities with the following sentence:

We reconstruct continuous-wave laser spectra, retrieve a line-shape width as small as 390 pm via fitting and resolve spectral peaks with a separation of 1.9 nm.

We re-formulated the sentence:

The full-width at half-maximum (FWHM, $\delta\lambda$) of the fitted curves is used as figure of merit for the quality of our designs and as a further quantification of the spectral resolution improvement. The obtained values for $\delta\lambda$ are 3.14 nm, 1.16 nm and 0.39 nm, respectively. Coefficients of determination (R^2) of the Lorentzian fitted curves to data-points increase from 0.93 to 0.99 for the three cases, which further confirms the improvement trend set by our framework.

into:

The full-width at half-maximum (FWHM, $\delta\lambda$) of the fitted curves is used as figure of merit for the quality of our designs and as a further quantification of the spectral resolution improvement. The obtained values for $\delta\lambda$ are 3.14 nm, 1.16 nm and 0.39 nm, respectively. Coefficients of determination (R^2) of the Lorentzian fitted curves to data-points increase from 0.93 to 0.99 for the three cases, which further confirms that the reconstructed spectra are approaching the actual CW laser line-shapes. It is important to highlight that the calculated FWHM should not be interpreted as the resolution of our devices since it is a quantity obtained via fitting of the data due to an insufficient number of points. The peak width could not be reliably calculated with just experimental points. It can, however, be used as a statistical measure to further testify the improvement trend set by our framework.

to better explain this aspect in the hope that it has now become clearer, thank you for highlighting this.

Incidentally, there was a rounding error when calculating the expected theoretical resolutions which, using our design parameters ($\lambda = 1550$ nm, $OPD = 267$ μ m, $n_{eff} = 1.931$) are approximated to

(including an additional significant digit) 4.66, 2.33, 1.16 nm, respectively. We changed the values in the main text as well.

Comment: *b) For the bandwidth, only two data points within 325nm are provided, which is inadequate in my viewpoint. Please provide more experimental results within 325nm optical range. Also, please plot the FWHM of the resolved peaks at different wavelengths, to see whether the spectrometer performs constantly within its bandwidth.*

Response: Thank you for highlighting this aspect as it helps us to better describe the performance of our device. We have run additional measurements and reconstructed the CW laser spectra of our tunable telecom sources in steps of 25 nm from 1460 nm to 1560 nm. Figure 5 is a modified version of the previous Fig. 5 in the original manuscript (which we updated now), where we have included these new measurements to show that our device performs consistently over the measurable bandwidth. The central panel at the bottom of the figure illustrates the retrieved FWHM as a function of wavelength, together with a linear fit indicated by the dashed line. The reason why we fitted the FWHM trend linearly is that we compared the trend obtained with our device to our commercial OSA reference spectra, as we observed a slight increase in FWHM while tuning the laser. The comparison between the two trends is now provided in the Supplementary Material (SM4, figure SF7, reported below in Fig. 6 for completeness). Even though we observe slight fluctuations in the reconstructed FWHM, which we once again attribute to the random nature of the DC drift effect, we believe it is clear that our device performs consistently over the measurable bandwidth and we hope that this clarifies the reviewer's doubt.

Figure 5: Experimental measurement of our device bandwidth, obtained by coupling CW laser lines at 1310 and 1635 nm and applying the standard spectral reconstruction framework. We observe a minimum measured bandwidth of 325 nm, although the expected operating window is larger as measurements were limited by equipment availability. The central bottom panel illustrates the reconstructed FWHM, obtained by fitting of the data, as a function of wavelength together with a linear fit of the data-points (solid line), to be compared with the reference FWHM measured with a commercial OSA (see Supplementary Material). This figure has replaced Fig. 5 in the manuscript.

We have reworked the corresponding discussion on pages 5-6 of the manuscript and extended it. The former paragraph said:

We experimentally test the operating range by coupling CW laser lines at 1310 and 1635 nm and apply the same framework as described above to resolve the spectra and estimate the bandwidth.

Figure 6: Comparison of FWHM for the tested CW laser spectra as reconstructed with our devices (left panel, solid line indicates a linear fit to the data) and measured with a commercial OSA (right panel).

We successfully reconstruct the spectral features and thus measure a device bandwidth of at least 325 nm as displayed in Fig. 5, where the insets illustrate the single laser lines fitted again to a Lorentzian line-shape, giving FWHM of $\delta\lambda = 0.39$ and 0.63 nm, respectively with R^2 values of 0.95 and 0.98.

while we have now re-written it as:

We experimentally test the operating range by coupling CW laser lines between 1310 and 1635 nm and apply the same framework as described above to resolve the spectra and estimate the bandwidth. We successfully reconstruct the spectral features and thus measure a device bandwidth of at least 325 nm as displayed in Fig. 5, where two bottom panels show zoom-ins of the two extreme laser spectra fitted again to a Lorentzian line-shape, giving FWHM of $\delta\lambda = 0.39$ and 0.63 nm, respectively with R^2 values of 0.95 and 0.98. Furthermore, to prove that our devices perform consistently across the measurable bandwidth, we sweep a tunable telecom laser in steps of 25 nm between 1460 nm and 1560 nm and reconstruct the spectra. We could retrieve each spectrum and followed the same data analysis framework described so far to retrieve the FWHM via fitting. The bottom central panel displays $\delta\lambda$ as a function of wavelength, together with a linear fit (dashed line) to the data. Fluctuations in the experimental values are to be attributed again to the random nature of the DC drift effect, which affects width and position of the spectrum if not perfectly compensated for. The linear fit is reported as inspection of the reference spectra revealed an increasing trend of the FWHM at the sources themselves, and a comparison between measured and reference values is reported in the Supplementary Material SM4. Despite our devices overestimate the line-shape width by around one order of magnitude, the performance is consistent across the measurable bandwidth and this further confirms the validity of our designs and framework.

in the hope that this analysis is now more convincing.

REVIEWER 3

Comment: *The manuscript presents an important optimization development of miniaturized spectrometers based on the SWIFTS concept using the EO effect in LiNbO₃. A few years ago, the group has reported the main concept Ref. 21, where the optical path difference (OPD) can be adjusted by applying a voltage to the LiNbO₃ layer. A larger OPD is desired for higher spectral resolution, and continuous interferogram scanning is required to increase the bandwidth of the spectrometer. A commercially available InGaAs camera (640 × 512 pixels) serves as the photodetection system in the experiments described. In Ref.21, the authors symmetrically recorded the emitted light from one line of the samplers. In this study, the authors design the optical waveguide in a meander fashion and recorded only the necessary half of the interferogram. In this way, a longer OPD can be recorded with the camera, resulting in a 5-fold increase in spectral resolution reaching 1 nm reflected in the title of the manuscript. The results are presented very clearly and the data are of a very high quality.*

Response: First of all, we would like to thank the reviewer for the positive evaluation of the manuscript, below is a point-by-point response to the comments.

Comment: *Could authors more explicitly state what is the limiting factor for the spectral resolution in the reported spectrometer? Is it the resolution of the camera 640 × 512 pixels capturing only the area of 267×214 micrometers? Or is it the DC-drift in the EO effect?*

Response: From a theoretical point of view, the only limitation to the achievable resolution is given by the finite length of the sampled interferogram (observable OPD), which in our paper we extend by splitting, delaying and recombining the signal in order to gain more information within the available FOV of the imaging system.

From a practical point of view, however, both the mentioned aspects have an impact and affect the resolution in a different way:

1. The field of view of the camera (which is limited by the objective magnification, 50x, needed to properly resolve and distinguish neighbouring nanowires) reduces the observable OPD and results in a lower resolution. More (and smaller) pixels could help but would need to be combined with a lower objective magnification, in order to effectively observe a longer interferogram portion per each scattering section. Eventually, our goal would be to integrate an InGaAs pixel array directly on top of the scattering region, allowing us to extend it further (we are currently discussing with companies on what kind of custom arrays are available). Therefore, in a truly integrated device there would get some gain on this aspect as we would remove the objective and have dedicated pixels to capture the pattern scattered by each nanowire.
2. The DC drift effect introduces chirping on the interferogram, which needs to be compensated for if the effect cannot be eliminated. Compensation for a CW laser is quite straightforward, as the interferogram is just a sinusoidal wave, yet for arbitrary spectra manifesting themselves in complicated interferograms with multiple frequencies, prior knowledge on the waveform is missing, thus compensation is complicated, data-processing becomes much more involved and stitching errors from combining interferograms at different applied voltages introduce artefacts in the reconstructed spectrum. This becomes more and more relevant as the spectral features to resolve get closer in frequency space, hence the DC drift artefacts effectively limit the achievable resolution by causing neighbouring features to merge during reconstruction (similar principle as for the Rayleigh criterion in imaging).

We have added the sentence, in the conclusion:

From an experimental point of view, the resolution of our devices is currently limited by two factors: FOV of the experimental setup and DC drift effects in lithium niobate. The restricted FOV of the imaging system (number of pixels and need of a large objective magnification) limits the observable OPD and thus the achievable resolution, while drifting of the EO response causes neighbouring spectral features to merge when reconstructing the spectrum and thus prevents us

from resolving some spectral shapes, especially when features are separated by a short spectral distance.

hoping that this aspect has now become clearer, thank you for pointing it out.

Comment: *Could authors comment on the difference between the theoretically expected spectral resolution values (4.8, 2.4, and 1.2 nm) and experimentally measured FWHM values (3.14, 1.16, and 0.39 nm).*

Response: The difference comes from the fact that the FWHM is a measure obtained through fitting of the experimental data to a Lorentzian model (we assume a CW laser, same model applied in our commercial OSA) and it does not correspond to the device resolution. Datasets do not include points at half the peak height (nor are they available at the same height), therefore it is not possible to use them to calculate the FWHM consistently and fitting is required. Via fitting, the spectral shape becomes closer and closer to a Lorentzian line-shape as the number of points increases since the reconstructed spectrum converges to the actual one (reasonably speaking, a delta function). This explains why the FWHM is reduced by a factor higher than two when comparing devices that should feature such an increase in spectral resolution, and is further confirmed by the R^2 value that increases consistently (0.926, 0.947, 0.989).

The smallest width that can be calculated from data-points is the distance between the points to the left and right with respect to the peak, and as can be seen from Fig. 7 (screenshot from the MATLAB tool), such distances match quite closely the expected theoretical resolutions (respectively 4.02, 2.12 and 1.3 nm). Considering the actual CW laser spectrum as a delta function, the FWHM obtained by actual experimental data-points (and not by fitting them) would be a good indicator of the device resolution, yet this is not possible due to lack of points and forces us to fit the data to get an estimate. If one looks at the distances between experimental data-points, these slightly differ from the calculated theoretical resolutions because of the following reasons:

1. The theoretical resolutions are calculated for the central wavelength of 1550 nm;
2. Points are not equally spaced in the reciprocal space as they are only in real space;
3. The calculations require to make an assumption on the effective mode index, which we obtain from simulations but it may be slightly different as it is highly dependent on the actual cross-section and therefore is affected by fabrication tolerances (and exact material properties).

To conclude, we reported the theoretical resolutions as calculated via equation (1) in the manuscript as it is the only quantity that can be directly related to design parameters. Using the distance between points in the reconstructed spectrum is not a reliable approach as, in principle, one could increase their density by applying a spline interpolation to the data. To further confirm that our designs introduce a clear performance improvement for our devices, we used a fitting approach to perform a statistical analysis on the reconstructed datasets. However, the FWHM obtained by the fit should not be interpreted as the devices resolution but just a figure of merit to confirm the validity of our approach.

We re-formulated the sentence:

The full-width at half-maximum (FWHM, $\delta\lambda$) of the fitted curves is used as figure of merit for the quality of our designs and as a further quantification of the spectral resolution improvement. The obtained values for $\delta\lambda$ are 3.14 nm, 1.16 nm and 0.39 nm, respectively. Coefficients of determination (R^2) of the Lorentzian fitted curves to data-points increase from 0.93 to 0.99 for the three cases, which further confirms the improvement trend set by our framework.

into:

The full-width at half-maximum (FWHM, $\delta\lambda$) of the fitted curves is used as figure of merit for the quality of our designs and as a further quantification of the spectral resolution improvement. The

Figure 7: Reconstructed CW laser spectra with highlights on the data-points that could be used to calculate the line-shape widths.

obtained values for $\delta\lambda$ are 3.14 nm, 1.16 nm and 0.39 nm, respectively. Coefficients of determination (R^2) of the Lorentzian fitted curves to data-points increase from 0.93 to 0.99 for the three cases, which further confirms that the reconstructed spectra are approaching the actual CW laser line-shapes. It is important to highlight that the calculated FWHM should not be interpreted as the resolution of our devices since it is a quantity obtained via fitting of the data due to an insufficient number of points. The peak width could not be reliably calculated with just experimental points. It can, however, be used as a statistical measure to further testify the improvement trend set by our framework.

to better explain this aspect in the hope that it has now become clearer, thank you for highlighting this.

Incidentally, there was a rounding error when calculating the expected theoretical resolutions which, using our design parameters ($\lambda = 1550$ nm, $OPD = 267$ μm , $n_{eff} = 1.931$) are approximated to (including an additional significant digit) 4.66, 2.33, 1.16 nm, respectively. We changed the values in the main text as well.

Comment: *Perhaps, the authors could state a scaling coefficient that links spectral resolution with the size of the spectrometer and another coefficient that links the size of the EO device to the OPD tunability range.*

Response: Thank you for pointing this out, as it is an interesting aspect to discuss. Fig. 8 illustrates (screenshot from the layout editor) the recombination regions for three device configurations featuring spectral resolutions of $\Delta\lambda = 2.33$, 1.16 and 0.58 nm, respectively. The latter has not been fabricated yet but follows the same working principle, with further delay of the waveform. The rectangular black frames surrounding the circuits show measurements of the physical circuit dimensions, in μm . The footprints are calculated as $1588 \times 760 \mu\text{m}^2 = 1.2 \text{ mm}^2$, $1815 \times 1322 \mu\text{m}^2 = 2.4 \text{ mm}^2$ and $2666 \times 1949 \mu\text{m}^2 = 5.2 \text{ mm}^2$, respectively. It is not trivial to find a simple scaling coefficient that links footprint of the sampling region to the resolution improvement as the growth in size is nonlinear due to the presence of additional Y-splitters. As can be seen from the calculations, between the first two configurations the area is doubled and corresponds to a factor of 2 improvement in resolution; between second and third configuration (additional factor of 2 improvement in resolution) the area is slightly more than doubled due to the necessity of splitting the optical power further. If one neglects the additional Y-splitters and

considers a fixed width for the looping circuits (for instance 1 mm, that can easily accommodate up to 12 parallel scattering regions with a $20\ \mu\text{m}$ spacing in the horizontal direction), the size increase in the vertical direction is of $800\ \mu\text{m}$ per each pair of new scattering regions introduced. For example, starting from the second configuration:

- Two scattering regions: $1000 \times 1815\ \mu\text{m}^2 = 1.82\ \text{mm}^2 \rightarrow \Delta\lambda = 1.16\ \text{nm}$
- Four scattering regions: $1000 \times 2615\ \mu\text{m}^2 = 2.62\ \text{mm}^2 \rightarrow \Delta\lambda = 0.58\ \text{nm}$
- Six scattering regions: $1000 \times 3415\ \mu\text{m}^2 = 3.42\ \text{mm}^2 \rightarrow \Delta\lambda = 0.39\ \text{nm}$
- Eight scattering regions: $1000 \times 4215\ \mu\text{m}^2 = 4.22\ \text{mm}^2 \rightarrow \Delta\lambda = 0.29\ \text{nm}$
- and so on...

From this calculation one could say that for an improvement factor of 2 in spectral resolution, the sampling region footprint grows roughly by 50%, even though there still is a slightly nonlinear trend. Considering instead the entire device with a fixed electrode length, the footprint increases by less than 30% for an improvement factor of 2 in spectral resolution. Additionally, the current designs are not optimised to minimise footprint but rather to eliminate as much as possible sources of inconsistency or difficulties in the data processing (e.g. bending losses or dispersive effects); delay lines can surely be implemented differently, and Y-splitters can be compacted to reduce the circuit area further.

Figure 8: Scattering region highlight from GDS designs of our spectrometer devices as taken from the layout editor. The rectangular boxes surrounding the circuit show their physical dimension in μm . From left to right, the expected theoretical resolutions are $\Delta\lambda = 2.33$, 1.16 and 0.58 nm, respectively.

We added the following sentence in the conclusion:

Concerning the latter aspect, at a fixed modulation region length (1 cm in this case, that determines the required stitching voltage in a linear fashion), the footprint of our devices increases by roughly 30% for each resolution improvement factor of 2. Currently, a modulation length of 1 cm and a spectral resolution of $\Delta\lambda = 1.16\ \text{nm}$ correspond to a device extending over a $20\ \text{mm}^2$ area. The current design is however not optimised for compactness but rather to minimise potential issues due to bending loss and dispersion; further circuit design engineering would allow to reduce the device footprint further.

to clarify these aspects, thank you for pointing it out.

Comment: *How realistic is solving the DC-drift problem by switching to the MHz operation regime? In this case the camera also need to capture images with MHz frame rate. This would require very fast cameras and large intensities of light.*

Response: Reaching MHz operation is indeed no trivial task, and with the current experimental setup and framework it is probably not possible due to integration time of the camera and time required to capture and save the images, due to the low light intensity and variability in pixel responsivity across the spectrum. However, few hundreds of kHz would anyway suit the purpose, and integration of a custom array of InGaAs pixels micro-positioned directly on top of the scattering region would allow to gain orders of magnitude in image brightness at the detectors, which would translate in a drastic reduction in integration time. We report in Fig. 4 an example IR image obtained with our setup during testing of our scattering nanowires (raw form); the image was obtained with an integration time of $10\ \mu\text{s}$ at $1550\ \text{nm}$ (single direction of propagation, used to estimate the scattering efficiency). As can be seen from the image, at some wavelengths and with correct setup configuration the integration time can be rather short, indeed $10\ \mu\text{s}$ would correspond to a modulation frequency of $100\ \text{kHz}$ at which no DC drift effect should occur. For reference, this image was obtained with an input optical power of $10\ \text{dBm}$, assuming a coupling loss of $7\ \text{dB}$ and neglecting propagation loss ($0.2\ \text{dB/cm}$) the waveguided (and scattered out) power is estimated to $3\ \text{dBm}$ ($2\ \text{mW}$). Due to optical losses in our imaging system, the collected power surely differs from the waveguided power, and this is why integration of a detector array would provide great advantage.

Figure 9: Raw IR image obtained during testing of our scattering nanowires with an integration time of $10\ \mu\text{s}$ at $1550\ \text{nm}$, estimated waveguided optical power is $3\ \text{dBm}$.

Moreover, further engineering of the scattering nanowires would allow to gain additional brightness as the device efficiency is not yet optimal (as discussed in the Supplementary Material), thus we believe that reaching data collection frequencies that would eliminate DC drift effects is possible.

REVIEWER 4

Comment: *This work demonstrated stationary-wave integrated Fourier-transform spectrometers (SWIFTSs) using monolithic thin-film lithium niobate.*

My major concern is the novelty of this work. The same group published a very similar work on Nature Photonics before in 2020. Similarly electro-optic effect in LN had been used to overcome the bandwidth limitations imposed by undersampling. The major difference between this work and the NP work is that this work switched from hybrid thin-film lithium niobate–silicon nitride (LN-SiN) platform to lithium niobate-on-insulator. Besides, minor structural changes had been made e.g. asymmetric region configuration and usage of Y-splitter.

This work is incremental and seems not suitable for NC.

Response: We understand the reviewer's point of view and thank them for their constructive criticism. However, we would like to bring up to their attention that the reported novelties are actually not the key points of our work, which we would summarise as follows:

1. The most important and original result we present is related to a new and smart approach to drastically improve the resolution of the spectrometer within a small footprint. Splitting and delaying the waveform with delay arms can be applied to all SWIFTSs systems and it allows to (virtually) indefinitely increase the resolution of the device while maintaining a very similar footprint. This is not just a minor structural change but it allows to break the resolution/bandwidth trade-off barrier that affects any integrated spectrometer. Currently, SWIFTSs devices can reach sub-nm resolution only by employing cm-long sampling regions. Our devices have potential to reach a spectral resolution of $\Delta\lambda = 290 \text{ pm}$ with a sampling region footprint of $1000 \times 4215 \text{ }\mu\text{m}^2 = 4.2 \text{ mm}^2$ by means of eight parallel scattering regions. The length of the modulation region depends only on the target voltage range that can be applied with the driving electronics, hence the device can be very compact if relatively large modulation voltages can be afforded.
2. The monolithic implementation on thin-film lithium niobate allows to easily combine the device with many other integrated active and passive building blocks already available on the platform. This allows to realize, with a single lithography and etching step for the optical layer, compact and functional circuits, hence expanding the applications of LNOI technology and adding an important building block to the portfolio. Integrated spectrometers in LNOI have not been reported yet and we believe that this is an important addition to the available circuits on the platform. Incidentally, the absence of a SiN ridge greatly reduces coupling losses to the device (avoids mode transition between materials) and lowers propagation losses by a factor of approximately 7 (1.4 dB/cm vs. 0.2 dB/cm).
3. The fabrication is fully lithographic, and on a wafer scale can be performed with deep-UV lithography. Employing modern (but commercially available) DUV systems would even allow to pattern the nanowires with the same lithography tool, hence fully enabling large-scale production at low cost as there is no need of FIB deposition any more.

We hope that the advantages provided by our framework have now become clearer and would be happy to further discuss otherwise.

-
- [1] Mario C. M. Souza, Andrew Grieco, Newton C. Frateschi, and Yeshaiahu Fainman. Fourier transform spectrometer on silicon with thermo-optic non-linearity and dispersion correction. *Nature Communications*, 9(1):665, December 2018. ISSN 2041-1723. doi: 10.1038/s41467-018-03004-6.
 - [2] Ke Zhang, Wenzhao Sun, Yikun Chen, Hanke Feng, Yiwen Zhang, Zhaoxi Chen, and Cheng Wang. A power-efficient integrated lithium niobate electro-optic comb generator. *Communications Physics*, 6(1):1–8, January 2023. ISSN 2399-3650. doi: 10.1038/s42005-023-01137-9.
 - [3] David Pohl, Marc Reig Escalé, Mohammad Madi, Fabian Kaufmann, Peter Brotzer, Anton Sergeev, Benedikt Guldemann, Philippe Giaccari, Edoardo Alberti, Urs Meier, and Rachel Grange. An integrated broadband spectrometer on thin-film lithium niobate. *Nature Photonics*, 14(1):24–29, January 2020. ISSN 1749-4885, 1749-4893. doi: 10.1038/s41566-019-0529-9.
 - [4] Amirhassan Shams-Ansari, Guan hao Huang, Lingyan He, Zihan Li, Jeffrey Holzgrafe, Marc Jankowski, Mikhail Churaev, Prashanta Kharel, Rebecca Cheng, Di Zhu, Neil Sinclair, Boris Desiatov, Mian Zhang, Tobias J. Kippenberg, and Marko Lončar. Reduced material loss in thin-film lithium niobate waveguides. *APL Photonics*, 7(8):081301, August 2022. ISSN 2378-0967. doi: 10.1063/5.0095146.
 - [5] Fabian Kaufmann, Giovanni Finco, Andreas Maeder, and Rachel Grange. Redeposition-free inductively-coupled plasma etching of lithium niobate for integrated photonics. *Nanophotonics*, 12(8):1601–1611, April 2023. ISSN 2192-8614. doi: 10.1515/nanoph-2022-0676.
 - [6] Mikhail Churaev, Rui Ning Wang, Annina Riedhauser, Viacheslav Snigirev, Terence Blésin, Charles Möhl, Miles H. Anderson, Anat Siddharth, Yuri Popoff, Ute Drechsler, Daniele Caimi, Simon Hönl, Johann Riemensberger, Junqiu Liu, Paul Seidler, and Tobias J. Kippenberg. A heterogeneously integrated lithium niobate-on-silicon nitride photonic platform. *Nature Communications*, 14(1):3499, June 2023. ISSN 2041-1723. doi: 10.1038/s41467-023-39047-7.
 - [7] Long Zhang, Shihan Hong, Yi Wang, Hao Yan, Yiwei Xie, Tangnan Chen, Ming Zhang, Zejie Yu, Yaocheng Shi, Liu Liu, and Daoxin Dai. Ultralow-Loss Silicon Photonics beyond the Singlemode Regime. *Laser & Photonics Reviews*, 16(4):2100292, 2022. ISSN 1863-8899. doi: 10.1002/lpor.202100292.
 - [8] Yan Gao, Fuchuan Lei, Marcello Girardi, Zhichao Ye, Raphaël Van Laer, Victor Torres-Company, and Jochen Schröder. Compact lithium niobate microring resonators in the ultrahigh Q/V regime. *Optics Letters*, 48(15):3949–3952, August 2023. ISSN 1539-4794. doi: 10.1364/OL.496336.

REVIEWERS' COMMENTS

Reviewer #1 (Remarks to the Author):

I appreciate the authors' detailed replies. I have no other questions.

Reviewer #2 (Remarks to the Author):

I have no further comments. The authors have clearly answered the questions by the reviewers.

Reviewer #3 (Remarks to the Author):

This reviewer would like to thank the authors for their careful response to all the questions and recommends the manuscript for publication in its current form.

Reviewer #4 (Remarks to the Author):

Thanks for clarifying the novelty. I would suggest acceptance now.